# Sex Differences in Visceral Pain and Comorbidities: Clinical Outcomes, Preclinical Models, and Cellular and Molecular Mechanisms

**DOI:** 10.3390/cells13100834

**Published:** 2024-05-14

**Authors:** Namrata Tiwari, Liya Y. Qiao

**Affiliations:** 1Department of Physiology and Biophysics, Virginia Commonwealth University School of Medicine, Richmond, VA 23298, USA; 2Department of Internal Medicine, Virginia Commonwealth University School of Medicine, Richmond, VA 23298, USA

**Keywords:** pain, visceral, sex, clinical, preclinical, molecule

## Abstract

Sexual dimorphism of visceral pain has been documented in clinics and experimental animal models. Aside from hormones, emerging evidence suggests the sex-differential intrinsic neural regulation of pain generation and maintenance. According to the International Association for the Study of Pain (IASP) and the American College of Gastroenterology (ACG), up to 25% of the population have visceral pain at any one time, and in the United States 10–15 percent of adults suffer from irritable bowel syndrome (IBS). Here we examine the preclinical and clinical evidence of sex differences in visceral pain focusing on IBS, other forms of bowel dysfunction and IBS-associated comorbidities. We summarize preclinical animal models that provide a means to investigate the underlying molecular mechanisms in the sexual dimorphism of visceral pain. Neurons and nonneuronal cells (glia and immune cells) in the peripheral and central nervous systems, and the communication of gut microbiota and neural systems all contribute to sex-dependent nociception and nociplasticity in visceral painful signal processing. Emotion is another factor in pain perception and appears to have sexual dimorphism.

## 1. Introduction

Defined by The International Association for the Study of Pain (IASP), chronic pain syndromes are classified into seven categories: chronic primary pain, chronic cancer pain, chronic post-traumatic and post-surgical pain, chronic neuropathic pain, chronic headache and orofacial pain, chronic visceral pain, and chronic musculoskeletal pain [1]. Patients with chronic visceral pain and irritable bowel syndrome (IBS) also often experience comorbidities including psychiatric disorders and somatic pain [2,3,4]. Visceral and somatic pain derived from numerous factors such as inflammation, injury, and/or metabolic disorders are unpleasant experiences in human and animals that greatly affect wellbeing and reduce quality of life. The sexual dimorphism of visceral pain and comorbidities has been documented extensively to be influenced by an intricate interplay of biological, psychological, and social factors [5,6,7,8,9,10,11,12,13,14,15,16]. Historically, the concept of sexual dimorphism in pain was largely based on clinical observations and cultural perceptions. The rise of gender equity and gender-specific medicine marks a pivotal shift in pain research, especially the women’s health movement, which catalyzes a deeper exploration into gender differences in health and disease including chronic visceral pain in a gender-based fashion [17,18,19,20].

In general, pain is categorized as nociceptive pain, neuropathic pain and nociplastic pain. Nociceptive pain, including visceral pain and somatic pain, is a type of pain that occurs when pain receptors in the tissues are activated by tissue inflammation or injury. Neuropathic pain is a result of nerve impairment. Nociplastic pain, on the other hand, is a functional pain disorder without apparent external tissue damage [21]. The generation and maintenance of pain involves the sensitization of primary afferent pathways including alterations of the activity of sensory neurons (nociceptors) and central sensitization, with the participation of glial and immune cells. The dorsal root ganglia (DRG) contain pain-sensing neurons to convey sensory signals from the peripheral organs (e.g., the pelvic organs and limbs) to the central nervous system (CNS). In general, DRG neurons are classified, in a broad category, into small, medium and large neurons, with the small neurons participating in the promotion of pain sensation. In visceral pain and associated comorbidities, changes in the neurochemical coding in sensory neurons and synaptic activity in the CNS, modulated by nonneuronal cells and organisms such as gut microbiota, are fundamental processes [22,23,24]. As for pain–sexual dimorphism, emerging evidence points out that molecular mechanisms other than the female sex hormones may have roles, since different stages of the estrus cycle have no greater impacts on nociception and analgesic responsiveness in females than those occurring intrinsically in males [25,26]. Recent progress derived from experimental animals sheds light on the role of sensory neuron subtypes and nonneuronal cells in visceral and somatic pain–sexual dimorphism as well as the sex-differential involvement of gut microbiota in chronic visceral pain and psychiatric comorbidities, such as depression and anxiety [27,28,29,30]. In this review, we examine the evidence of visceral pain–sexual dimorphism in humans, and discuss the experimental animal models that may or may not resemble or align with the nature of human bowel dysfunction. We summarize sex differences in sensory biological mechanisms in visceral pain, including but not limited to changes in nociceptors, glial and immune cells, gut microbiota regulation, and CNS plasticity. Moreover, visceral pain is accompanied by many comorbidities, such as psychiatric disorders and somatic pain, that involve central sensitization and neuroplasticity (Figure 1). Many of these aspects of research on mechanisms of visceral pain–sexual dimorphism are just emerging, but some evidence from somatic pain–sexual dimorphic studies can suggest the potential mechanisms in visceral pain. Most of the supporting evidence is derived from works of literature that have comparisons between sexes. Key words used for the literature search include, but are not limited to, sex, gender, male, female, pain, nociception, bowel, colon, visceral pain, somatic pain, dimorphism, etc., under different combinations. Both review and original research papers are cited. 

## 2. Clinical Findings in Visceral Pain–Sexual Dimorphism

A comparative study of colonic functionality in adult men and women illustrates a substantial effect of sex on the rectal discomfort threshold, in which noxious sigmoid stimulation causes the most sensitivity in women with IBS and the least sensitivity in healthy women when compared to men with respective health conditions [5]. No significant differences in rectal discomfort thresholds are seen between healthy men and men with IBS [5], inferring less sensitive perception of noxious stimulation in men with or without IBS. However, a recent study where 280 young healthy men and women (equal numbers) undergo rectal balloon distensions to measure visceral sensory thresholds and pain shows no sex differences nor IBS-related risk factors [31]. Another study comprising an equal ratio of males and females in IBS and healthy groups (70% and 72% are women in each group) demonstrates that patients with IBS have lower sensory thresholds to rectal distention compared with healthy subjects, whereas the threshold for first sensation is similar in both groups [32], appearing as though the sex composition in both groups does not skew the results. An internet-based nationwide survey of 88,607 people, including 44,815 males (50.6%) and 42,176 females (47.6%) in the US, to assess the prevalence of IBS and comorbidities in a 7 day recall period, covers a variety of pain modalities and symptoms, such as anal pain, belly pain or discomfort, bladder pain, bloating, bowel incontinence, constipation, diarrhea, difficulty swallowing, excess gas, heartburn or acid reflux/regurgitation, nausea or vomiting, pelvic pain, rectal pain, revealing that 6.1% met Rome IV IBS criteria, among which women (1.39–2.06 fold higher compared to men) and every day smokers (1.32–1.73 fold higher compared to nonsmokers) have higher odds for having all three IBS subtypes, i.e., IBS with mixed bowel habits (IBS-M), IBS with constipation (IBS-C) or IBS with diarrhea (IBS-D). Many comorbidities such as prior gastroenteritis, fibromyalgia, gallstones, peptic ulcer disease, and thyroid disorders are associated with increased odds for all three IBS subtypes, while diabetic people tend to have IBS-D and IBS-M but not IBS-C [33].

For disease progression, IBS-C is more prevalent in women than men, however, fewer men are enrolled in relevant clinical trials [34]. IBS-D appears more prevalent in men than women [8,35,36]. In the development of post-infectious (PI) IBS (IBS-PI), female sex, younger age, and psychological distress during infectious enteritis are risk factors [37]. An additional risk factor identified from a study of 231 patients (131 males and 100 females) with acute gastroenteritis is the longer duration of antibiotics use in IBS-PI [38]. In this study, the percentages of those developing IBS-PI assessed by phone interview are 10.7% (14 out of 131) males and 20% (20 out of 100) females [38]. However, two IBS patient cohorts with 231 patients in Rome II and 141 patients in Rome III category who undergo rectal barostat testing in combination with questionnaires for anxiety, depression, somatization, and abuse, demonstrate that the pain threshold in Rome II cohort is positively associated with female gender, while pain threshold in Rome III cohort is positively associated with male gender [39]. The conflicting results are affected by a variety of factors such as hormonal levels altered by aging, emotional and psychological aspects of pain perception, early-life adversity, availability of medical care, diet selections, societal roles, tendency to report, and geographical and cultural differences [11,34,40,41]. As for the role of sex hormones in visceral pain and pain at large, numerous original research and review articles have acknowledged them [15,42,43,44,45]; however, the specific modulatory effects of sex hormones in pain whether as the initiators, exacerbators, or endogenous protectors require a more detailed investigation. Transgender patients who have pain-related conditions such as headaches, fibromyalgia, temporomandibular (TMD) myalgia, and visceral pain have provided additional information on the role of hormones and the effectiveness of hormone replacement therapy in pain management [46], which will not be further addressed here.

Physicians’ attitudes and practices also play a role in identifying and grading the severity of visceral pain [47]. This is no different from the assessment of other pain modalities in which the gender biases of the participants and testers are apparent [12,48,49,50]. For example, women have higher pain ratings tested by men while men report less pain when tested by women. Imbalanced social status and peer pressure between participants and experimenters also affect the outcomes of pain ratings. In addition, terms such as “tough” boys and “brave” men occur in pain tests [12,50]. A meta-analysis of sex differences in healthy children between 0 and 18 years old in response to experimental pain, encompassing various pain-inducing tasks like cold pressor, heat, and pressure pain, demonstrates that there are no significant differences between boys and girls on pain-related outcomes from the majority of studies, however, some studies show that girls have higher cold pressor pain intensity and lower heat pain threshold than boys, but not mechanical detection and pain stimuli [51,52]. A 3 year follow-up interview on a cohort consisting of 95 females and 48 males, aged 5–23 years, with initially reported pain, shows that all genders report continuing pain, with females significantly more likely than males to report, especially for patients whose pain is associated with psychosocial factors [53]. Interestingly, in a study comprising 73 children (37 boys, 36 girls) of 4–12 years old undergoing cold pressor pain tasks monitored by either their father or mother, fathers give their sons higher pain ratings than their daughters, whereas mothers’ ratings do not differ among boys and girls [48]. Although visceral pain and general pain are different, they share similar pathways in pain perception and involve complex sensory and emotional experience. Hitherto, the sexual dimorphism of visceral pain and pain at large in humans has been documented with contradictory outcomes [12,50], which is likely due to a variety of psychosocial and cultural elements [11].

## 3. Visceral Pain Preclinical Models with Sexual Dimorphism

Understanding the underlying molecular mechanisms in visceral pain sexual dimorphism is crucial for identifying key molecules/pathways for potential drug targets. Experimental animal models have been developed rigorously to resemble the phenomenon seen in humans. Colonic inflammation has been used to recapitulate inflammatory bowel disease (IBD)-associated bowel dysfunction and post-inflammatory visceral pain, one subtype of IBS (i.e., IBS-PI). Early-life inflammation or stress, as well as adult stress, disturbance of gut microbiota, and genetic models are also implemented to study the mechanisms of IBS (Table 1).

It appears that visceral pain is easier to develop in male mice as a result of colonic inflammation or enema treatment with organic chemicals. Zymosan-induced behavioral–visceral hypersensitivity is detected in male mice but not female mice, which is accompanied by a higher number of mechanosensitive colorectal afferents per mouse following the zymosan treatment of male mice but not female mice [54]. A single dose of intracolonic 2, 4, 6-trinitrobenzenesulfonic acid (TNBS) instillation induces colitis in both sexes on day 3, with male mice more severe than female mice [55]. By day 7 following TNBS treatment, colonic hypersensitivity is more pronounced in male mice than in female mice, which is mediated by Piezo2 in nociceptive neurons [27]. Similarly, dextran sodium sulfate (DSS) treatment to induce colonic inflammation affects male mice more strongly than female mice showing less inflammatory infiltrates, less crypt damage, and lower TNFα production in the colon of female mice than male mice [56], with sex-differential changes in the immune responses in the colon [57]. Histamine enema increases visceral hypersensitivity in male mice but not female mice and deletion of histamine N-methyltransferase (HNMT) from enteric glia protects males from histamine-driven visceral hypersensitivity, with no effects on female mice [58]. The male-preferred colonic inflammatory pain (nociceptive pain) in experimental mice is similar to that observed in clinics showing IBS-D to be more prevalent in men than women [35]. 

In rats, the sex differences in visceral pain are strain- and disease model-dependent. For example, female rats tend to develop visceral hypersensitivity following unpredictable odor attachment learning [59,60], serotonin transporter (SERT) knockout [61,62], or repeated water avoidance stress (rWAS) [63]. While maternal separation causes visceral hypersensitivity in male but not female Long Evans rats [59], and in both male and female Wistar rats [64,65]. The protocols of maternal separation also make a difference, by which type P (separation of half of littermates) induces visceral hypersensitivity preferably in Wistar female but not male rats [64]. Additionally, visceral sensitivity is significantly increased in female Long Evans rats exposed to unpredictable early-life odor-shock presentation when compared to female rats from the predictable early-life stress or odor-only group, which is not seen in male rats [60]. Female but not male SERT KO rats demonstrate a higher action potential of colonic afferent neurons compared to wildtype rats of the same sex [61], suggesting a female-predominant role of the serotonin system in the development of IBS. This notion is further confirmed by the application of 5-HT3 receptor antagonists or agonist showing that 5-HT3 receptor signaling is essential in visceral hypersensitivity in female SERT-KO rats [62]. As for rWAS (1 h stress daily for a consecutive 4 days), visceral motor responses (VMR) is increased in female rats but not male rats in response to 60 mmHg colorectal distension (CRD) examined 24 h post rWAS [63]. A study in rat TMD disorder that has IBS comorbidity also shows stronger and longer-lasting visceral pain behaviors in female rats than male rats [66]. These rat models with stress are valuable tools in investigating the molecular mechanisms of sex-dependent non-inflammation-related nociplastic visceral pain. 

Gut microbiota communicates with nociceptors to affect pain perception, also in a sex-dependent fashion. For example, the instillation of a bacterially derived short-chain fatty acid Isovalerate into the gut increases visceral hypersensitivity in male mice but not female mice [29]. In female germ-free mice, a lack of microbiota does not affect visceral sensitivity [67], which is in stark contrast to germ-free male mice in which visceral sensitivity is higher than conventional control mice [23]. Similarly, early-life antibiotic treatment of rats results in visceral hypersensitivity in males but not females [68], suggesting a sex-differential role of microbiota in affecting gut pain which is male predominant. This could relate to a stronger microbiota–gut–brain interaction in males [69] (also see later section on gut–brain axis). Interestingly, the perturbation of microbiota by antibiotics induces visceral hypersensitivity in female mice [70], and also attenuates intraperitoneal acetic acid- or intracolonic administration of capsaicin-produced acute pain in female mice [71]. Another study shows that the absence of gut microbiota increases visceral sensitivity to CRD in both male and female mice [72]. The discrepancy of sex-differential outcomes in visceral pain affected by microbiota may lie in the spectrum of antibiotics used in experiments [68] and/or diverse metabolomic responses [24]. 

The etiology of visceral pain (inflammation vs. non-inflammation) appears to relate to pain–sexual dimorphism; however, species (rats vs. mice) also seem to have a correlation in which male mice and female rats are more prone to disease development (Table 1) although stringent parallel comparisons have not been made. Some other examples with exceptions may add more information into the models (Table 1). A number of studies show that colonic sensitivity in naïve animals is higher in females than males when comparing within the same strain, regardless of the species being rats [73,74,75] or mice [27,76], while other studies show opposite observations, that naïve male mice (C57Bl/6j) respond more strongly to CRD than female mice [77], similar to guinea pigs [78]. An additional study shows no difference in colonic sensitivity between naïve male and female rats [63]. Mustard oil treatment (induce colonic inflammation) induces visceral pain in both male rats and female rats with female rats having a greater increase than male rats [73]. TNBS treatment of guinea pigs evokes visceral hypersensitivity in both sexes with males being stronger (male 7.6- vs. female 2.5-fold increase) [78]. These results suggest that visceral pain–sexual dimorphism in experimental animals could be species- and strain-dependent, and could also be affected by the protocols used to induce and assess pain. Limited experiments suggest that female rats, compared to male rats, more easily develop visceral hypersensitivity, contrary to mice or guinea pigs in which visceral pain often occurs or is stronger in males than females. Other findings in rats and mice show different sexual dimorphic changes in visceral inflammation and pain, compared to the above categories (Table 1). In response to the induction of bladder inflammation, female rats and male rats show similar changes in the urinary bladder [79]. Both female and male rats demonstrate visceral hypersensitivity after early-life maternal separation [80]. Another study combining neonatal maternal separation with early-life TNBS colitis in rats shows that male adult rats have hyperactivity behaviors while female adult rats that have undergone the same procedure did not show abnormalities in behavioral tests [81]. Maternal separation (nociplastic pain) also induces visceral hypersensitivity in male mice, in which female mice are not assessed [82].
cells-13-00834-t001_Table 1Table 1Sex-differential outcomes of colonic inflammation, hypersensitivity and comorbidities. VH: visceral hypersensitivity; M: male; F: female; MS: maternal separation.ModelSpeciesSexFindingsReferences**Male predominant**



zymosanmiceM, FVH in M but not F[54]TNBSmiceM, Finflammation (day 3) and VH (day 7) more severe in M than F[27,55]DSSmiceM, Finflammation and immune response in colon more severe in M than F[16,56,57] histamine enemamiceM, FVH in M but not F[58]neonatal MS +TNBSratsM, Fadult hyperactivity in M but not F[81]baselineC57Bl/6jM, Fcolonic sensitvity M > F[77]**Female predominant**



early life stressratsM, FVH in F but not M[59,60,64]SERT KOratsM, FVH in F but not M[61]TMDratsM, Fstronger and longer-lasting VH in F than M[66]baselinerats, miceM, Fcolonic sensitvity F > M[27,73,74,75,76]**Both sexes**



mustard oilratsM, Fboth sexes have VH, F > M[73]MSratsM, Fboth sexes have VH at adult[64,65,80]Microbiota



IsovaleratemiceM, FVH in M but not F[29]germ freemiceM, FVH in male but not F[23,67]early life antibioticsratsM, FVH in male but not F[68]antibioticsmiceFVH, analgesic on acute pain[70,71]germ freemiceM, FVH in both sexes[72]**Referred pain**



colitisratsM, Fboth have bladder hyperactivity[83,84,85,86,87,88]colitismice, ratsMbladder and somatic pain[89,90,91,92]DSSmiceM, Freferred pain comparable in M and F, F shows stronger acute pain and licking and freezing behavour [16]

## 4. Sex Differences in Visceral Pain Comorbidities

### 4.1. Clinical Evidence

Visceral pain comorbidities occur in both women and men with different etiologies. One of the major comorbidities associated with visceral pain is urinary tract (UT) dysfunction. Studies in female patients, exploring the associations between lower UT and gastrointestinal symptoms, show that nocturia is significantly more prevalent in patients with fecal incontinence [93]. Female patients with functional constipation or IBS also report urinary frequency and incontinence, nocturia, straining to void, or incomplete voiding [93,94,95]. Although constipation is less prevalent in men than in women [96], men with constipation-like bowel habits (three or fewer bowel movements per week) are also associated with nocturia, incomplete bladder emptying, and urinary hesitancy [97]. However, men with more than 10 bowel movements per week (diarrhea-like), experience nocturia only [97]. In children, constipation (9.4% of boys and 12.4% of girls) results in 6.8 times more occurrences of urinary tract dysfunction than non-constipation [98]. As for bladder-to-bowel crosstalk, both men and women with an overactive bladder report chronic constipation or fecal incontinence with a similar percentage [96]. Girls and boys seen at the urology clinic for voiding dysfunction and at the GI clinic for functional constipation also have similar levels of symptoms [99]. In a population who have adrenoleukodystrophy, the prevalence of urinary and bowel symptoms is similar in men (75.0%) and women (78.8%), however, the onset of the bladder and bowel disease occurs a decade earlier in men [100]. These findings suggest that bowel abnormality is associated with bladder dysfunction in both sexes regardless of age. 

Patients with IBS also demonstrate comorbid chronic somatic pain response to pressure, heat, or a cold pressor applied to upper extremities (neck, back and shoulder), with a higher percentage in females than in males (54.1% vs. 39.5%) [101]. Sex differences are more pronounced in cold-pressure tolerance, cold pressor intensity rating, but not pressure pain threshold [101]. Nonabdominal chronic pain in adolescents (mean age 16.1 years old) with IBS occurs more often in girls than in boys (22.6% vs. 13.5%), and nearly one third of the IBS cases report nonabdominal chronic pain [102]. Another pronounced comorbidity of visceral pain is mental disorders, which also shows a sex difference (see review [103]). Microbiota plays a key role in the gut–brain axis that underlies visceral pain-associated psychiatric disorders and neurodegenerative diseases, with sex differences [104]. The association of gut microbiota and somatic pain, migraines, and autoimmune disorders is also documented in patients [105,106,107,108], suggesting a variety of pain comorbidities affected by gut microbiota that have critical roles in visceral pain. 

### 4.2. Preclinical Studies

Animal models to study visceral pain comorbidities mainly use experimental rodents (Table 1). In rats, normal females, males, or ovariectomized females with colitis demonstrate bladder hyperactivity [83,84,85,86,87,88]. Male rats are also used to study prostate-to-bladder cross-sensitization [109]. In mice, males that have colitis are detected for bladder hypersensitivity, and vice versa, male mice that have bladder inflammation exhibit colonic hypersensitivity [89,90,91,92]. Male mice are also used to assess the effects of colitis on somatic and psychological disorders [110,111]. In addition, male colitic rats and mice have been evaluated to show somatic, mechanical and thermal pain [91,112,113,114,115], and increased calcium (Ca^2+^) activity in paw innervating afferent neurons in male mice [115]. Colitis-induced pain comorbidity in female mice has not been rigorously examined, which could be due to the lesser effectiveness of inducing colonic inflammation and colonic hypersensitivity in female mice [27,29,54,56]. One of the studies using DSS colitic model demonstrates that referred pain post DSS inflammation is comparable in male mice and female mice, except that female mice have more pronounced licking and freezing behaviors than male mice [16]. We assessed hind paw mechanical sensitivity in female mice following TNBS-induced colitis and compared it to male mice, which showed that the onset of the referred mechanical pain in female mice was delayed when compared to that of male mice (Figure 2). On day 7, following colitis induction, female mice did not exhibit robust colonic mechanical hypersensitivity [27] but showed strong referred pain (Figure 2). Similarly, DSS induces lesser colonic inflammation in female mice than in male mice [56]; however, the referred pain sensitivity in both sexes is similar following DSS treatment [16]. One of the explanations could be that the colon is the primary organ injured in colitis, while the hind paws are not directly injured. The underlying mechanisms of the differential behaviors of the injured vs. uninjured organs in female mice are not yet understood. This necessitates a comparison of the neurochemical coding between the injured primary afferent neurons and the uninjured neurons. 

## 5. Cellular and Molecular Mechanisms in Sexual Dimorphism of Visceral Pain and Pain at Large

Evidence in sexual dimorphism of visceral pain and pain at large has inspired rigorous studies to uncover the underlying molecular mechanisms to suggest drug targets for sex-oriented effective treatment. Sensory neurons are of primary importance for this type of study, and nonneuronal cells also spark considerable research interest. Neural circuits and neurotransmitters are integral components in transducing pain signals; thus, peripheral damage can be timely sensed and remedied. Sexual dimorphic molecular mechanisms in pain have been found at all levels in the pain circuits. A recent study shows that the brain-derived neurotrophic factor (BDNF), a visceral and somatic pain modulator, downregulates inhibitory signaling elements and upregulates excitatory elements in the spinal cords of rodents and human tissues of males but not of females [116], shedding light on the central mechanisms in pain processing in a sex-dependent manner. There are some exciting but few results on the molecular basis of visceral pain–sexual dimorphism, while the rapid progress in this aspect, in somatic pain and pain at large, provides valuable information in understanding the general molecular process in the pain circuits that could be translated to visceral pain.

### 5.1. Transcriptomic Analysis of DRG

Comparison of male and female DRGs from mice reveals 540 sex-dependent differentially expressing genes (DEG) in DRG sensory neurons [117]. Specifically, female mice express a distinct set of genes predominantly linked to immune processes, nervous system development, extracellular matrix organization, and inflammatory pathways. These genes include those related to the complement system, chemokines, cytokines, and growth factor receptors, with a more prominent expression in trigeminal ganglion than DRG neurons [117]. Male mice demonstrate a different profile of predominantly expressed genes, focusing on proteasome subunits, mitochondrial function, and oxidative phosphorylation [117]. These findings point to potential protective mechanisms in male sensory neurons against nerve injury, highlighting sex-specific resilience pathways. Pain-related genes in the DRG of male and female mice, such as those related to transcriptional and translational machinery, are also different [117]. Further comparison of the molecular profiles of sensory neurons between L6-S1 DRG (the majority of visceral sensory neurons) and L4-L5 DRG (the majority of somatic sensory neurons) uncovers 466 genes different between lumbar and sacral DRGs, which are characteristic of markers for large, myelinated sensory neurons such as Nefh, Pvalb, Ntrk3, Runx3, etc., in the lumbar DRG and nociceptor transducers such as Trpv1, preprotachykinin (Tac1), Pacap, etc., in the sacral DRG [118]. Important sex-differential gene expression in the sacral DRG is also identified. For example, preproenkephalin (Penk), that competes with and mimics the effects of opiate drugs, has a higher expression level in adult male sacral DRG and Trpv1 neurons than that in female mice [118]. While Kdm6a, a X-linked gene that acts as histone demethylase, has a higher expression level in female sacral DRG neurons as well as in Trpv1 neurons than that in male mice [118]. As for other pain modalities, an earlier study demonstrates that there are no striking differences in gene expression between male and female mice, while immune cell infiltration into DRG is sexually different in nerve injury pain [119]. A number of other studies [120,121,122,123,124,125] have also provided considerable amounts of datasets from RNAseq to reveal sex-differential changes in molecular signatures in DRG, the spinal cord, sciatic nerves, and brain tissues from mice and rats in pain, which are summarized in Table 2, to provide guidance for an in-depth investigation of pain–sexual dimorphism in the visceral organs and other tissues.

RNAseq analysis of DRGs from more than 50 patients for the first time reveals profound sex-differential molecular signatures in pain and provides more objective measurements at the molecular levels. In brief, male DRG samples have higher expression levels of IL1B, Tnf, CXCL14, Osm, EgrB, TrpV4, Lif, CCL3/4 and female DRG have higher levels of CCL1, CCL19, CCL21, Penk, TrpA1, adenosine A2B receptor and glycine receptor alpha 3 in pain when compared to non-pain controls [126]. These factors are closely related to pain, nociception or inflammation [127]. Strikingly, the commonly upregulated genes associated with pain in both sexes are only a handful [126], suggesting an intrinsic sex-differential sensory biological process in pain. Of note, 80% of the subjects are older than 55 years [126], which suggests that these sex-dependent changes in pain-associated molecules are unlikely to be the results of sex hormonal fluctuations in females due to the hormonal decline in older populations.

### 5.2. Nociceptors

Calcitonin gene-related peptide (CGRP) and Nav1.8 are largely co-localized in nociceptive neurons to sense pain signals [128,129]. CGRP is produced in DRG neurons and can release peripherally to promote neurogenic inflammation or centrally to cause spinal central sensitization. Following colitis induction in male rats, CGRP is increased in DRG neurons, primarily in uninjured bladder afferent neurons and also in the spinal cord [130,131]. A CGRP antagonist, CGRP 8–37, attenuates colitis or nerve growth factor (NGF)-induced visceral hypersensitivity in male rats [132]. When chemogenetically activating peripheral glia, that results in visceral and somatic hypersensitivity in male but not female mice, CGRP expression levels are also increased in DRG neurons of male mice but not female mice [133]. In an acute arthritis pain model or cyclophosphamide-induced bladder pain, CGRP participates in peripheral and spinal sensitization and pain in male rats [134,135]. Following nerve injury, blockage of the CGRP receptor inhibits neuropathic pain behaviors in rats of both sexes [136]. CGRP is also implicated in migraine pathophysiology that largely affects women. An anti-CGRP antibody can attenuate colitis-induced bladder hypersensitivity in female rats [86]. In hyperalgesic priming induced by activation of interleukin 6 signaling, an intrathecal injection of CGRP receptor antagonists olcegepant or CGRP8-37 blocks and reverses hyperalgesic priming and neuropathic pain only in female mice but not male mice, and a CGRP-sequestering monoclonal antibody blocks IL-6-induced mechanical hypersensitivity and hyperalgesic priming in female mice and rats [137]. These findings suggest that the sex-differential role of CGRP in pain is context-dependent and animal model-specific. 

The sex-differential function of the Nav1.8 ion channel is also suggested by recent studies. Optogenetic activation of Nav1.8-lineage sensory fibers elicits nerve fiber firing much easier in female mice than male mice at baseline [29]. Conditional deletion of Piezo2 from Nav1.8-lineage neurons reduces baseline colonic mechanical sensitivity in female mice but not male mice [27]. These findings suggest that female mice may have a narrower sensory homeostatic range at baseline (weaker inhibitory feedback when nociceptive neurons are activated and less strength to maintain activity when nociceptors are inhibited) than male mice. During pain states, Piezo2 conditional deletion from nociceptors attenuates TNBS-induced colonic hypersensitivity in male mice but not female mice [27], has no impact on complete Freund’s adjuvant (CFA)-induced knee swelling in female mice [28], but reduces knee swelling associated with knee hyperalgesia in female mice, noting that male mice in this CFA context are not measured [28]. Conversely by sex, nociceptors lacking Piezo2 are associated with reduced experimental osteoarthritis-induced joint pain and referred mechanical allodynia of the hind paw, as well as NGF-induced knee swelling and hyperalgesia in male mice, when compared to control [28]. Spared nerve injury (SNI) induces mechanical hypersensitivity and cold allodynia similarly in both male and female mice; however, female mice demonstrate higher expression levels of ATF3 and HMGB1 expression in Nav1.8-lineage nociceptors than male mice [138]. Interestingly, Nav1.8-Cre based Toll-like receptor 4 (TLR4) deletion attenuates acutely (day 1–5 post SNI) but not chronically (day 7 post SNI) the mechanical hypersensitivity in female mice, but has no effects on mechanical pain in male mice; conversely, TLR4 conditional deletion reduces cold allodynia up to day 7 post SNI as measured in male mice but not in female mice [138]. It appears that the sex-differential roles of nociceptors in the regulation of pain are diverse by measured pain modalities and the time course examined after injury or stress. Visceral pain and somatic pain could be processed differently, as well as the mechanisms underlying inflammatory nociceptive pain, neuropathic pain, nociplastic pain, or referred pain and pain comorbidities.

### 5.3. Nonneuronal Cells

Microglia in the spinal cord are mostly studied for their roles in pain in a sex-dependent manner. In visceral pain, inhibition of microglia by minocycline, MAC-1-conjugated saponin (a method of depleting microglia), in female rats reduces visceral pain comorbidity [87]. Nerve injury induces mechanical allodynia in both male and female mice; however, the inhibition of microglia through pharmacological approaches or chemogenetics reduces the development of the neuropathic pain in male mice but not female mice [139,140,141]. Conditional deletion of Orai1 channel from microglia attenuates microglia development in the spinal cord and mitigates pain hyperalgesia following nerve injury in male mice but not in female mice [142]. Chemogenetic activation of microglia elicits mechanical allodynia exclusively in male mice, although the reactive microglia-dominant molecules are upregulated in microglia in both sexes [143]. The male-predominant role of microglia in neuropathic pain is conserved in rats and mice [144]; this is different from visceral pain where microglia have a role in female rats [87]. The female-preferential role of microglia in rats is also shown by a study in which lipopolysaccharide (LPS) treatment induces greater microglia activation and interleukin IL-1β upregulation in the periaqueductal gray of female rats compared with male rats [145]. The male-predominant role of microglia in mice with pain is likely due to sex-differential development of microglia by which male mice have delayed gene expression relative to females, and the exposure of adult male mice to LPS accelerates microglial development in male mice [146]. Astrocytes, on the other hand, regulate inflammatory and neuropathic pain in a sex-independent manner in mice [147]. Satellite glial cells (SGCs) compose the largest glial populations in DRG and mediate inflammation-induced mechanical allodynia in male mice [133,148]. Chemogenetic activation of SGCs inhibits acute (15 min to 1 h after clozapine N-oxide (CNO) treatment) mechanical hyperalgesia in male mice (female mice are not examined) [149], and facilitates chronic (1–3 days with daily CNO injection) mechanical pain and colonic hypersensitivity in male mice but not female mice [133]. Strikingly, the number of Proteolipid protein Plp1-expressing SGCs in DRG of female mice is significantly less than those in male mice [133]. RNAseq information comparing DRG neurons and immune cells in DRG and the spinal cord shows that female and male mice develop nerve injury pain with no apparent changes in gene-expression profiles in DRG neurons and in the amount of immune cell activation in the spinal cord, but with sex differences in immune cell infiltrations to DRG [119]. 

The non-hormonal molecular differences between sexes are apparent, which is rapidly developing with a vast amount of new information being discovered. Table 3 lists some examples of cells and molecules that have roles in the sexual dimorphism of visceral pain and pain at large, which can be expanded along with the progress in research in this direction. Of note, findings in this area only provide limited information and have not achieved consistent results, likely due to the animal species and models used, diverse methodologies and approaches, and the limited robustness in power analysis for sex differences. Additional information is needed to pinpoint the non-hormonal molecular profiles in visceral pain and pain at large, therefore guiding personalized, gender-specific diagnosis and treatment.

## 6. Sexual Dimorphisms of Gut–Brain Axis and Emotional Effects on Visceral Pain and Comorbidity

One of the major comorbidities of visceral pain is psychiatric disorders (Figure 1). Modification of the gut–brain axis plays a critical role in this process. In patients with IBS and healthy subjects, emotion-related cognitive processes are observed with sex differences [150,151]. In mice, perturbation of the activity of enterochromaffin (EC) cells not only affects visceral pain but also regulates anxiety-like behaviors, in a sex-dependent manner [29]. Gut microbiota are tightly correlated with abnormal brain activity and mental disorders such as depression in both men and women [152]. These findings suggest a sex-differential process in the higher order of function in the nervous system in perceiving visceral pain and necessitate a sex-specific approach in understanding and managing these conditions. A couple of working models in the gut–brain connectome have been proposed recently by Mayer et al., [2] that may involve the microbiome–immune loop, bidirectional microbiome–neuronal interaction, sensorimotor plasticity, and autonomic regulation, which manifests central sensitization in the IBS-associated psychiatric disorders. 

During moderate rectal inflation (physical stimuli) and anticipation of rectal inflation (psychological stimuli), women show greater activation in the ventromedial prefrontal cortex, right anterior cingulate cortex (ACC), and left amygdala than men [153]. The amygdala has a pivotal role in the experiencing of emotions and its activity is regulated by stress [154] in a sex-dependent manner, also identified in rats following prolonged stress challenge, which is shown as reduced cholinergic activity in the amygdala of female but not male mice [155]. In healthy subjects consisting of 16 men and 16 women who undergo esophageal distension at the pain toleration level, no sex differences are found in personality scores, anxiety levels, skin conductance response (SCR), or subjective ratings of pain. Although the pattern of brain activation and deactivation during anticipation and pain are similar between sexes, women and men show distinct levels of activation and deactivation in certain brain areas. For example, women show greater deactivation in the right amygdala and stronger limbic inhibition during anticipation, suggesting more emotional responses to painful stimuli when compared to men [156]. In a task-free resting-state functional magnetic resonance imaging analysis, female healthy subjects and IBS patients have a frequency power distribution skewed toward high frequency to a greater extent in the amygdala and hippocampus compared with male subjects [153]. At the molecular level, CGRP8-37 administration into the central nucleus of the amygdala of rats reduces neuropathic mechanical hypersensitivity in both sexes but shows female-predominant effects on emotional affective responses (ultrasonic vocalizations) and anxiety-like behaviors (open field test) [136]. In experimental rats that undergo early-life stress, which induces visceral hypersensitivity in female but not male adults, epigenetic regulation occurs in the central nucleus of the amygdala to modulate the expression of corticotrophin-releasing hormone (CRH), a critical factor in stress responses and anxiety [60,157], providing a molecular basis in the female-preferred emotionally related pain process. 

The connectivity between brain areas and other components in the pain pathway are implicated in the regulation of chronic pain. The connectivity includes but is not limited to microbiota–gut–brain connections [158,159], the functional interactions within the brain [160,161], and the connectivity of brain and the descending pain antinociceptive system [162]. Emerging evidence points out the ACC, a key brain area in pain processing, as a therapeutic target for central neuromodulation in clinical pain management [163,164,165]. A systemic review on the connection of gut microbiota and the activity of brain areas reveals that the insula and ACC are most frequently associated with the gut microbiota [159]. The role of ACC in visceral pain has been well documented in experimental animals; unfortunately, most of them focus on using male rodents. During visceral pain processing, glutamatergic neurons in the claustrum project to the ACC to regulate ACC neuronal activity in male mice; however, female mice are excluded from this study, thus no sex comparison is made [160], although sex differences have been found in the connectivity of ACC with other brain areas in the processing of chronic pain [162]. Specifically, women show more functional connectivity than men between the sgACC and the PAG, raphe nucleus, medial thalamus, and anterior midcingulate cortex (aMCC) [162], and women’s sgACC functional connectivity to the default mode and sensorimotor networks is higher than men’s [166]. In a cross-sectional study on resting-state functional connectivity (RSFC) among older adults, sex differences are also evident in the associations of thermal pain with RSFC between the ACC and amygdala, and between the ACC and PAG in older females compared to older males [167]. Moreover, using a 64-channel multielectrode (MED64) system to record synaptic plasticity in the ACC, a classical low-frequency stimulation (LFS, 1 Hz, 900 pulses) evokes long-term potentiation (LTP) with no sex-related difference, however, long-term depression (LTD) that weakens synaptic strength is higher in the ACC in male mice than in female mice [168]. The expression of GABAergic genes is lower in the ACC of men but higher in the ACC of women with mental disorders, such as Schizophrenia, that are often interconnected with chronic pain [169]. These studies suggest that women’s brain wiring may allow for higher engagement of the antinociceptive system to mediate pain habituation than men.

## 7. Concluding Remarks

There is a caveat in confirming sexual dimorphism of pain in humans, due to the interference of emotions, geographical and societal status, and testing environments. Objective measurements of visceral pain and other pain formats are necessary to decipher pain–sexual dimorphisms, which will benefit from the identification of pain markers. Adequate animal models can provide a means of research to understand the molecular and signaling processes in pain, with a hope to translate these to humans. Additionally, animal models provide more reproducible states to investigate the underlying sex-differential pain mechanisms. Mice and rats are popular experimental animals for the study of visceral pain and pain at large; however, the majority of studies have been focusing on using males. Limited research data using both sexes thus far suggest that male mice are prone to be inflamed in the colon and have visceral pain when compared to female mice. Female but not male rats are reported to have visceral pain under other types of pathophysiology, such as stress. It is certain that more experiments should be enforced to ensure that a sufficient body of information is generated to better understand sex differences in visceral pain. Sensory neurons, immune cells and central neural circuits, especially those involved in emotion, have sex differences at baseline and during diseases. However, how these evolving landscapes contribute to sex-differential pain generation and maintenance is still ambiguous and necessitates further investigations.

## Figures and Tables

**Figure 1 cells-13-00834-f001:**
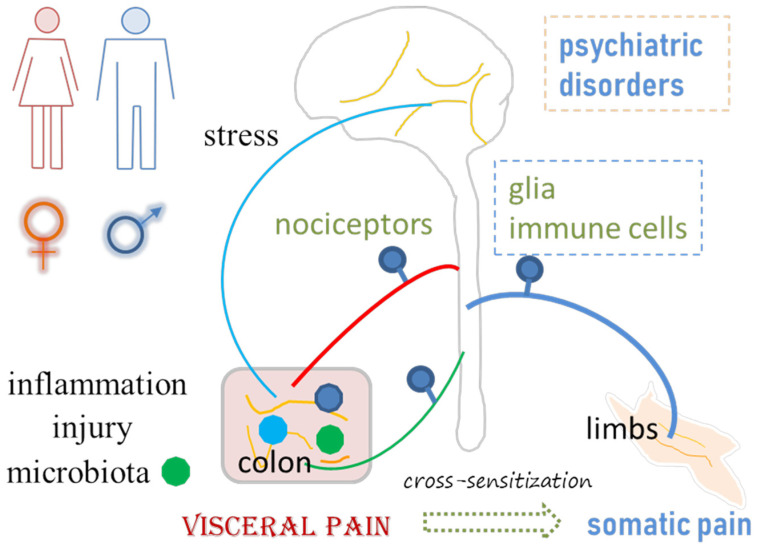
Visceral pain emanates from or is associated with inflammation, injury, gut microbiota imbalance, stress, psychiatric disorders, and somatic pain. Neuroplasticity in the peripheral and central nervous systems, including changes in the properties of neurons, glial cells and immune cells, plays a role in visceral pain and associated comorbidities.

**Figure 2 cells-13-00834-f002:**
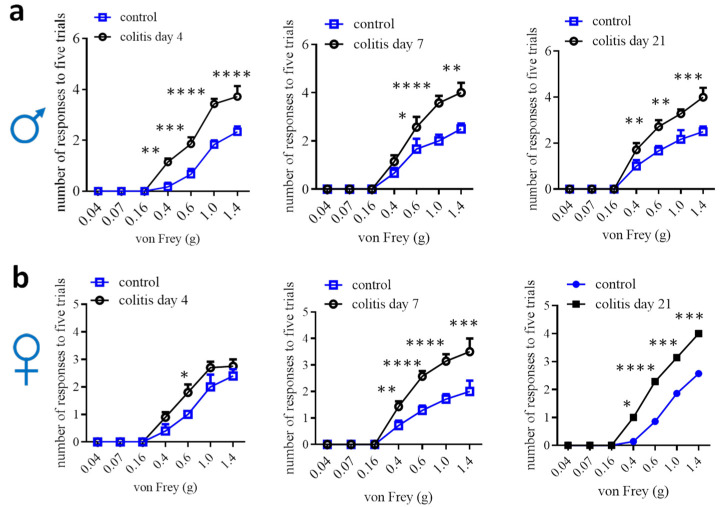
Comparison of hind paw mechanical sensitivity in male (**a**) and female (**b**) mice after TNBS-induced colitis. Male: control n = 6, colitis n = 7; female day 4: control n = 5; colitis n = 10; female day 7 and 21: control n = 7, colitis n = 7. Two-way ANOVA with Tukey’s multiple comparison test. *, *p* < 0.05; **, *p* < 0.01; ***, *p* < 0.001; ****, *p* < 0.0001.

**Table 2 cells-13-00834-t002:** Transcriptome analysis of tissues involved in pain processing to shed light on sex differences at molecular levels. SC: spinal cord; DRG: dorsal root ganglia; SN: sciatic nerves.

Pain Model	Species	Samples	Tissues	Sex Differences	Refs
SNL	mice	FACS	SC, DRG	immune cell infiltration	[119]
MBP	mice	tissue RNA	SN, DRG, SC	PLC/IP3R female, PI3K male	[120]
SNI	mice	microglia	SC	inflammatory microglia M > F	[121]
CCI, CIPN	mice	microglia	SC	reactive microglia M > F	[122]
SNI	rats	tissue RNA	DRG, SC	T-cells, immune responses, neuronal transmission and plasticity	[123]
SNI	mice, rats	tissue RNA	SC	gene downregulation only occurs in female	[124]
CCI	rats	tissue RNA	DRG	neuroinflammation in both sexes with different molecular changes	[125]

**Table 3 cells-13-00834-t003:** Examples of cellular and molecular regulation of visceral pain and pain at large in a sex-dependent manner.

Molecules and Cells	Sex	Pain Modalities	Refs
CGRP	male	visceral hypersensitivty	[130,131,132,133,135]
CGRP	both	neuropathic pain	[136]
CGRP	female	cytokine-induced pain; cross-organ sensitization	[86,137]
Nav1.8	F > M baseline	afferent excitibilty	[29]
Nav1.8-target Piezo2cKO	F > M baseline	colonic mechanosensing	[27]
Nav1.8-target Piezo2cKO	male	visceral hypersensitvity; joint pain	[27,28]
Nav1.8-target Piezo2cKO	female	inflamatory pain	[28]
Nav1.8-target TLR4cKO	female	nerve injury pain	[138]
Nav1.8-target TLR4cKO	male	cold allodynia	[138]
microglia	female	visceral pain, LPS treatment	[87,145]
microglia	male	neuropathic pain	[139,140,141,142,143,144,146]
astrocytes	sex independent	nerve injury pain; inflammatory pain	[147]
satellite glia	male	visceral pain; inflammatory pain	[133,148]

## Data Availability

All data are included in this paper.

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
