# Peer review of "Sex Differences in Visceral Pain and Comorbidities: Clinical Outcomes, Preclinical Models, and Cellular and Molecular Mechanisms"

_cells, 2024, doi:10.3390/cells13100834_

Round 1

Reviewer 1 Report (New Reviewer)

Comments and Suggestions for Authors

This is an interesting and timely review that addresses a very important theme.  It is well organized and well referenced, however the section on preclinical data related to sex difference and visceral sensitivity needs a little bit more work and revision.

 Major comments

1)  There are a number of studies which haven’t been referenced and the influence of sex and sex hormones in visceral pain whether basal, post inflammation or post stress is not as straight forward as the authors seem to suggest it.

For instance page 4, line 167 and page 5 line 224: Sex differences in the basal visceral pain response are not always female predominant, and are likely strain-dependent, as strain per se has a very major effect on the visceral pain response, but can also be related to the protocol of distension: See following papers

Kamp EH, Jones RC 3rd, Tillman SR, Gebhart GF. Quantitative assessment and characterization of visceral nociception and hyperalgesia in mice. Am J Physiol Gastrointest Liver Physiol. 2003 Mar;284(3):G434-44. doi: 10.1152/ajpgi.00324.2002. Epub 2002 Nov 20. PMID: 12444012.

Larsson M, Arvidsson S, Ekman C, Bayati A. A model for chronic quantitative studies of colorectal sensitivity using balloon distension in conscious mice -- effects of opioid receptor agonists. Neurogastroenterol Motil. 2003 Aug;15(4):371-81. doi: 10.1046/j.1365-2982.2003.00418.x. PMID: 12846725

Moloney RD, Dinan TG, Cryan JF. Strain-dependent variations in visceral sensitivity: relationship to stress, anxiety and spinal glutamate transporter expression. Genes Brain Behav. 2015 Apr;14(4):319-29. doi: 10.1111/gbb.12216. Epub 2015 Apr 21. PMID: 25851919.

Larauche M, Mulak A, Kim YS, Labus J, Million M, Taché Y. Visceral analgesia induced by acute and repeated water avoidance stress in rats: sex difference in opioid involvement. Neurogastroenterol Motil. 2012 Nov;24(11):1031-e547. doi: 10.1111/j.1365-2982.2012.01980.x. Epub 2012 Jul 9. PMID: 22776034; PMCID: PMC3470786.

 Bellucci F, Buéno L, Bugianesi R, Crea A, D'Aranno V, Meini S, Santicioli P, Tramontana M, Maggi CA. Gender-related differential effect of tachykinin NK2 receptor-mediated visceral hyperalgesia in guinea pig colon. Br J Pharmacol. 2016 Apr;173(8):1329-38. doi: 10.1111/bph.13427. Epub 2016 Mar 7. PMID: 26758701; PMCID: PMC4940818.

 Rosztóczy A, Fioramonti J, Jármay K, Barreau F, Wittmann T, Buéno L. Influence of sex and experimental protocol on the effect of maternal deprivation on rectal sensitivity to distension in the adult rat. Neurogastroenterol Motil. 2003 Dec;15(6):679-86. doi: 10.1046/j.1350-1925.2003.00451.x. PMID: 14651604.

 López-Gómez L, López-Tofiño Y, Abalo R. Dependency on sex and stimulus quality of nociceptive behavior in a conscious visceral pain rat model. Neurosci Lett. 2021 Feb 16;746:135667. doi: 10.1016/j.neulet.2021.135667. Epub 2021 Jan 22. PMID: 33493648.

Along the same line, page 5, line 185: early life adversity induces alterations of visceral pain in both sex, the female predominance is only seen in very specific circumstances and models. The authors also do not address the potential for strain differences in the sex differential responses to stress in those models of visceral hypersensitivity which are actually discussed and commented upon in one of these manuscript.

Holschneider DP, Guo Y, Mayer EA, Wang Z. Early life stress elicits visceral hyperalgesia and functional reorganization of pain circuits in adult rats. Neurobiol Stress. 2016 Jun 1;3:8-22. doi: 10.1016/j.ynstr.2015.12.003. PMID: 26751119; PMCID: PMC4700548.

 Prusator DK, Greenwood-Van Meerveld B. Sex-related differences in pain behaviors following three early life stress paradigms. Biol Sex Differ. 2016 Jun 10;7:29. doi: 10.1186/s13293-016-0082-x. PMID: 27293543; PMCID: PMC4901516.

 Rosztóczy A, Fioramonti J, Jármay K, Barreau F, Wittmann T, Buéno L. Influence of sex and experimental protocol on the effect of maternal deprivation on rectal sensitivity to distension in the adult rat. Neurogastroenterol Motil. 2003 Dec;15(6):679-86. doi: 10.1046/j.1350-1925.2003.00451.x. PMID: 14651604.

Lastly, the influence of estrous cycle and sex hormones on visceral pain is also not straight forward. Some studies do support a role of estrous in visceral pain responses, but others do not. Estrogen is generally believed to exacerbate visceral pain but note should be taken that other studies have shown that ovariectomy can actually lead to visceral hypersensitivity in mice. This should be more nuanced…

2) It is not clear to me why the use of opioids is brought in the manuscript in the abstract and the conclusion. If it is to highlight the fact that pain is sex dependent and that opioid exhibit sex-differential effects on the pain relief, it would then be worth discussing the preclinical data related to sex difference in the opioid effects on visceral sensitivity.

Minor comments

1) Page 3, line 103: Introduce the acronyms IBS-C, IBS-D and IBS-M before using them in the paragraph. They are detailed in the next paragraph, but should be with first use of acronym.

Page 4, line 163: replace subtypes by subtype.

Page 5, line 201: Replace “of noting” by “of note”

Comments on the Quality of English Language

Overall it’s well written, but may require editorial review for minor grammar error and misspellings.

Author Response

Thank you for your constructive suggestions to improve this paper. We have attached our point-by-point responses for your review.

Reviewer 2 Report (New Reviewer)

Comments and Suggestions for Authors

Sex Differences in Visceral Pain and Comorbidities: Clinical Outcomes, Preclinical Models, and Cellular and Molecular Mechanisms

Summary: The authors do an excellent job describing sex differences in the mechanisms underlying visceral pain and associated comorbidities. I think it’s a worthy manuscript for publication, but I have a few comments.

Major Points:

  1. “In the introduction, the authors state that peripheral nociceptor activation is a root cause of pain that are classified…” While it’s true that this process is extremely important in pain transmission, I don’t think this sentence is necessary here since pain can persist without nociceptor activation. That being said, I also think the authors can remove all the “The”s that are at the start of every sentence. This is not grammatically necessary. I think this reference may help the authors describe the different types of well more completely as well.

  1. Cohen SP, Vase L, Hooten WM. Chronic pain: an update on burden, best practices, and new advances. Lancet. 2021;397(10289):2082-2097. doi:10.1016/S0140-6736(21)00393-7

  1. The sentence in line 58 is a bit confusing. I can’t tell if the authors are referring to one specific female sex hormone, or female sex hormones in general. Please clarify. Moreover, I see many of these english/grammatical issues throughout the paper. For example, lines 68-70 are oddly worded. It would be much simpler to say “We summarize sex differences in sensory biological mechanisms in visceral pain…” Another example is in line 107. I think the authors meant to type “meta-analysis data”, not “meta-data analysis”. I think it would be crucial to go through the paper again and check for these english/grammatical issues, and get outside help if necessary.

  1. In line 121, the authors mention “psychological pain threshold”. It’s not completely clear what the authors mean here. Could they please elaborate?

  2. In line 488, the authors describe what the amygdala is involved with functionally speaking. While this is true, I think what they have currently is extremely reductive. The amygdala has been associated with emotions, learning, memory and attentional mechanisms, and is involved in multiple brain networks as it has many efferents throughout the brain. I think a couple sentences of primer would be very useful here.

  1. Roozendaal, B., McEwen, B. & Chattarji, S. Stress, memory and the amygdala. Nat Rev Neurosci 10, 423–433 (2009). https://doi.org/10.1038/nrn2651

  1. Mayer, E.A., Ryu, H.J. & Bhatt, R.R. The neurobiology of irritable bowel syndrome. Mol Psychiatry 28, 1451–1465 (2023). https://doi.org/10.1038/s41380-023-01972-w

  1. The first sentence of the concluding remarks doesn’t seem to have much direction, and I’m not sure what the purpose of it is. I suggest it be removed. Opioids can be useful clinically in some patients, and should not be discounted entirely. In the sentence regarding females require “much more” morphine, it would be useful to put a quantitative value here.

Minor Points:

  1. In line 108, it should be IBS-C, not C-IBS.

  2. In line 113, the authors say “adds additional risking factors”. This sounds 

  3. Line 249 should say “...Visceral Pain Comorbidities”

Comments on the Quality of English Language

Sex Differences in Visceral Pain and Comorbidities: Clinical Outcomes, Preclinical Models, and Cellular and Molecular Mechanisms

Summary: The authors do an excellent job describing sex differences in the mechanisms underlying visceral pain and associated comorbidities. I think it’s a worthy manuscript for publication, but I have a few comments.

Major Points:

  1. “In the introduction, the authors state that peripheral nociceptor activation is a root cause of pain that are classified…” While it’s true that this process is extremely important in pain transmission, I don’t think this sentence is necessary here since pain can persist without nociceptor activation. That being said, I also think the authors can remove all the “The”s that are at the start of every sentence. This is not grammatically necessary. I think this reference may help the authors describe the different types of well more completely as well.

  1. Cohen SP, Vase L, Hooten WM. Chronic pain: an update on burden, best practices, and new advances. Lancet. 2021;397(10289):2082-2097. doi:10.1016/S0140-6736(21)00393-7

  1. The sentence in line 58 is a bit confusing. I can’t tell if the authors are referring to one specific female sex hormone, or female sex hormones in general. Please clarify. Moreover, I see many of these english/grammatical issues throughout the paper. For example, lines 68-70 are oddly worded. It would be much simpler to say “We summarize sex differences in sensory biological mechanisms in visceral pain…” Another example is in line 107. I think the authors meant to type “meta-analysis data”, not “meta-data analysis”. I think it would be crucial to go through the paper again and check for these english/grammatical issues, and get outside help if necessary.

  1. In line 121, the authors mention “psychological pain threshold”. It’s not completely clear what the authors mean here. Could they please elaborate?

  2. In line 488, the authors describe what the amygdala is involved with functionally speaking. While this is true, I think what they have currently is extremely reductive. The amygdala has been associated with emotions, learning, memory and attentional mechanisms, and is involved in multiple brain networks as it has many efferents throughout the brain. I think a couple sentences of primer would be very useful here.

  1. Roozendaal, B., McEwen, B. & Chattarji, S. Stress, memory and the amygdala. Nat Rev Neurosci 10, 423–433 (2009). https://doi.org/10.1038/nrn2651

  1. Mayer, E.A., Ryu, H.J. & Bhatt, R.R. The neurobiology of irritable bowel syndrome. Mol Psychiatry 28, 1451–1465 (2023). https://doi.org/10.1038/s41380-023-01972-w

  1. The first sentence of the concluding remarks doesn’t seem to have much direction, and I’m not sure what the purpose of it is. I suggest it be removed. Opioids can be useful clinically in some patients, and should not be discounted entirely. In the sentence regarding females require “much more” morphine, it would be useful to put a quantitative value here.

Minor Points:

  1. In line 108, it should be IBS-C, not C-IBS.

  2. In line 113, the authors say “adds additional risking factors”. This sounds 

  3. Line 249 should say “...Visceral Pain Comorbidities”

Author Response

Thank you for your constructive comments to strengthen this manuscript. We've made point-by-point responses and revise the text accordingly. 

Round 2

Reviewer 2 Report (New Reviewer)

Comments and Suggestions for Authors

The authors have done a good job at addressing my concerns and have greatly improved the manuscript. I think it is suitable for publication.

Comments on the Quality of English Language

N/A

This manuscript is a resubmission of an earlier submission. The following is a list of the peer review reports and author responses from that submission.

Round 1

Reviewer 1 Report

Comments and Suggestions for Authors

Presumably, this is a well-written review article updating sex differences in visceral pain. However, I cannot find Tables 1 and 2 in the manuscript or the journal website. Thus, it is not easy to evaluate this manuscript as it is. Importantly, I recommend the authors add a section and/or table about “response to pain medication”. The authors briefly mentioned dosing issues and differences in SERT KO mice. Thus, the question is whether women prefer certain pain medications or whether they use or require different doses and duration of treatment depending on co-morbid or co-occurring disorders. In addition, it would be helpful if the authors included any illustrations to explain types of visceral pain and significant non-hormonal sex differences.

Comments on the Quality of English Language

appropriate

Author Response

Report Notes

Reviewer 2 Report

Comments and Suggestions for Authors

Thank you for the opportunity to review your manuscript。This article mainly reviews the gender differences in visceral pain and its comorbidities。These differences and their reasons are mainly introduced through three aspect:Clinical Outcomes, Preclinical Models,

and Cellular and Molecular Mechanisms。However, the main scientific issue in this paper is the difference between men and women in pain sensitivity. The cognition of pain is a relatively subjective concept, which is not only the severity of the physical disease, but also has a greater relationship with psychological and social factors. The evaluation of pain is limited to some pain rating scales, and there are no appropriate objective indicators. The author uses some animal models and some molecular mechanisms to elaborate his own views, but it is relatively messy and the idea of explanation is not clear. It is recommended to also make a table in the section on cellular and molecular mechanisms, listing the effects of different molecules up or down, present or absent, on the threshold of visceral pain in different sexes. And some of the comorbidities listed may also need to be more logical. To explain the effect of comorbidities on gender differences in pain perception.

Comments on the Quality of English Language

The quality of English in this paper is appropriate and there are no major grammatical problems.

Reviewer 3 Report

Comments and Suggestions for Authors

Tiwari and Qiao review the literature focusing on sex differences in visceral pain in clinical and animal models with special attention on IBS and other forms of bowel dysfunction and its comorbidities. Also, the authors review articles focused on animal models that try to describe cellular and molecular mechanisms underlying sexual dimorphism in visceral pain and other types of pain.

In general terms, the review is well-written and structured and it was easy for the reader to follow all the different sections of the manuscript.

However, on page 5, when the authors indicated some preclinical models of visceral pain in which microbiota plays a role, there are a few studies that point to sex and the oestrus cycle are critical biological variables affecting the influence of gut microbiota in visceral pain perception. Interestingly, male germ-free presents visceral hypersensitivity and this hyperalgesic phenotype is reversed by postnatal colonization with microbiota from conventional mice (Luczynski et al., 2017). However, germ-free female mice did not present visceral hypersensitivity and estrous-cycle-induced changes in visceral pain were abolished in the germ-free animals (Tramullas et al., 2021).

Furthermore, several studies using antibiotics to eradicate the gut microbiota in male rats (Verdú et al., 2006; O’Mahony et al., 2014; Aguilera et al., 2015; Hoban et al., 2016;) demonstrate that the absence of gut microbiota induces visceral hypersensitivity. However, the early-life treatment with antibiotics did not affect visceral sensitivity to colorectal distension in female animals (O’Mahony et al., 2014).

All these references are in line with the topic of the manuscript. I consider that it is reasonable to include them in the review, under the appropriate heading.

Minor comments:

There is a typo in the table 1.

Early lide instead of early life

 Without more comments, I believe the review in its present form is adequate to be published in Cells.

Reviewer 4 Report

Comments and Suggestions for Authors

The manuscript by Tiwari and Qiao aimed to address sexual dimorphisms in visceral pain and comorbidities and to summarize several current findings ranging from clinical outcomes to molecular mechanisms. While I generally appreciate the effort undertaken in this endeavor, this review would greatly benefit from a much stronger focus and a more concise approach and, in the current form, was not really a pleasure to evaluate.

Criteria for including literature are completely unclear. The focus on visceral pain is widely lost starting already in the introduction, comorbidities, despite being mentioned prominently in the title, are not systematically considered, and the work is extensive, yet for the most part lacks a proper structure. It remains elusive as to why the authors so often broaden their scope to pain in general and even beyond. To name only few of several examples for the lacking “guiding thread” throughout the work, the authors describe the role of testing environment using the relevance of investigator’s sex / gender in subjective pain reports, to then proceed to prevalence rates of IBS subtypes, then emphasize cultural factors and the societal impact of the opioid crisis with no reasonable context to conclude the relevance of elucidating biological underpinnings to overcome biases. In another section, the authors even include findings on schizophrenia in the context of ACC, a brain region for which numerous animal and human findings in health and chronic visceral pain conditions exist, out of which, conversely, several have been ignored.

Comments on the Quality of English Language

No comments regarding the quality of english language.